# A Ti-6Al-4V Milling Force Prediction Model Based on the Taylor Factor Model and Microstructure Evolution of the Milling Surface

**DOI:** 10.3390/mi13101618

**Published:** 2022-09-27

**Authors:** Siyuan Zhu, Man Zhao, Jian Mao, Steven Y. Liang

**Affiliations:** 1School of Mechanical and Automotive Engineering, Shanghai University of Engineering Science, Shanghai 201620, China; 2School of Mechanical Engineering, Georgia Institute of Technology, North Ave NW, Atlanta, GA 30332, USA

**Keywords:** Ti-6Al-4V, milling force, Taylor factor, EBSD, microstructure

## Abstract

In this paper, a milling force prediction model considering the Taylor factor is established, and the Ti-6Al-4V milling force predicted by the model under different milling parameters is presented. In the study, the milling experiment of Ti-6Al-4V was carried out, the milling force was collected by the dynamometer, and the microstructure evolution of the milling surface before and after milling was observed by EBSD. Through the comparative analysis of the experimental results and the model prediction results, the reliability of the prediction model proposed in this study was verified, and the influences of the milling parameters on the milling force were further analyzed. Finally, based on the EBSD observation results, the effects of the milling parameters on the microstructure evolution of the milling surface were studied. The results show that both the tangential milling force and normal milling force increase with the increase in the milling depth and feed rate. Among the milling parameters selected in this study, the milling depth has the greatest influence on the milling force. The average errors of the tangential milling force and normal milling force predicted by the milling force model are less than 10%, indicating that the milling force prediction model established in this paper considering Taylor factor is suitable for the prediction of the Ti-6Al-4V milling force. With the change in the milling parameters, the grain structure, grain size, grain boundary distribution, phase distribution, and micro-texture of the material surface change to varying degrees, and the plastic deformation of the milling surface is largely coordinated by the slip.

## 1. Introduction

Titanium is an allotrope, and α-titanium has an HCP (hexagonal close-packed) structure, while β-titanium has a BCC (body-centered cubic) structure. Titanium alloy is an alloy based on titanium and is composed of other elements. Common titanium alloys are divided into the following three categories: α-titanium alloy, β-titanium alloy, and (α+β) titanium alloy [1]. Ti-6Al-4V is a typical (α+β) titanium alloy, and Ti-6Al-4V is also one of the materials widely used in the field of aerospace engineering [2,3,4]. Because Ti-6Al-4V has the physical properties of a high hardness, high strength, and low thermal conductivity, it produces a very high temperature in the cutting process, and the material is easily attached to the tool surface. At present, there are certain common problems regarding the machining process of titanium alloy, such as a low machining efficiency and serious tool wear. In the field of aerospace engineering, the material removal per unit time in the actual processing of titanium alloy workpieces is much less than that of other materials. Therefore, the question of how to select a reasonable processing method and improve the machining efficiency and quality has become an important research topic [5].

The cutting force is a very important parameter in the machining process, which has direct impacts on the stability and efficiency of the machining process and further affects the quality of the machined surface and tool wear. Over the last century, many scholars have carried out a great deal of research on the prediction of the cutting force, and their research methods can be divided into the empirical model, finite element model, and analytical model [6]. Based on the machining principle, the analytical method uses geometric and mechanical methods to analyze the cutting process, which can reflect the real machining state and predict the results accurately. The classical analytical model includes the right angle cutting model proposed by Merchant [7] and Oxley [8], which assumes that the shear plane is located at the place where the shear energy is minimal and calculates the shear angle according to this assumption. Based on the classical model, Endres et al. [9] and Moufki et al. [10] introduced the ploughing force model and force–thermal coupling model, respectively, which further improved the prediction accuracy of the analytical model. By using the combination of the analytical model and finite element simulation, Yang [11] established a constitutive model considering the large deformation, large strain rate, and high temperature characteristics and predicted the cutting force and cutting deformation in the milling process of titanium alloy. Das et al. [12] proposed a new method of grey relational analysis of the Ti6Al4V machining parameters based on Taguchi. The effects of the cutting speed, tool feed, and cutting depth on the matching response were studied. Kim [13] carried out the milling force and thermal analysis of Ti-6Al-4V by the finite element method, predicted the temperature distribution, and evaluated the surface roughness, tool wear, and cutting force. Compared with the results of traditional face milling, laser-assisted face milling improves the surface roughness. In addition, in terms of tool damage, laser-assisted face milling has been demonstrated to be more effective than traditional face milling. Perez et al. [14] proposed a Taylor model based on oblique cutting so as to quantify the effect of the crystal structure on the shear strength. The microstructure, crystal texture, and particle morphology of different samples were analyzed by EBSD, and the tool geometry, tool position, and laser scanning path were taken into account. The analysis showed that there is a significant interaction between the plane direction of the shear band and the grain orientation on the spindle, and there is a strong correlation between the cutting force and the prediction model. Based on the J–C (Johnson–Cook) flow stress model and mechanical–thermal coupling model, Ji [15] established a cutting force prediction model considering the cutting parameters, tool parameters, material properties, and surface lubrication parameters. Through the coupled iterative calculation method, this model iteratively determines the cutting force and cutting heat in the cutting process and is better than other models in terms of the scope of its application and prediction accuracy.

The above studies on the prediction models of the cutting force do not account for the microstructure of the material. In the process of material deformation, the evolution of the microstructure usually has an important influence on the physical properties of the material [16]. Many scholars use EBSD (electron backscattering diffraction) to analyze the microstructure changes in materials (including the microstructure, grain size, slip, twin, etc.). Based on the new method of external friction theory, Tong et al. [17] analyzed the friction-reducing and anti-wear mechanisms of micro-texture cutting tools. The optimum area ratio of the micro-texture in the contact zone between the chip and tool was studied when the cutting force was considered. Su et al. [18] used PCD tools to study the effect of ultrasonic vibration-assisted machining (UVAM) on the machinability of the Ti6Al4V titanium alloy and its mechanism. The cutting force, machined surface, surface bonding force, and wear morphology were analyzed. The results showed that UVAM can effectively improve the cutting performance. Wu et al. [19] carried out a comparative experiment on the dry milling of the Ti-6Al-4V alloy with micro-texture cutters and common cutters. The results showed that the micro-texture tool can reduce the cutting force, cutting temperature, and power consumption by about 15%, 10%, and 5%, respectively. The results showed that the cutting radial width, cutting speed, and axial cutting depth have obvious influences on the dry grinding of the Ti-6Al-4V alloy, and the mechanism for improving the performance of the microstructure tool is mainly its self-lubrication. Dabwan et al. [20] studied the effects of the feed speed, radial cutting depth, and cutting speed on the surface roughness, cutting force, microhardness, microstructure, chip morphology, and surface morphology of Ti6Al4V. It was found that different orientations have different effects on the machined surface. The results showed that the EBM parts machined in the TLP direction have a good surface quality and surface integrity. To sum up, the microstructure of the material has an important influence on the mechanical properties, and it also affects the cutting force and cutting temperature in the material processing.

In the above studies, the effects of the grain orientation on the properties of the materials were not considered. In this paper, the Taylor factor model is introduced, and the Taylor factor is taken as a variable. Based on the milling force prediction model proposed by Ji [15], an improved milling force prediction model considering the effect of the grain orientation is established. Combined with the milling parameters of a small titanium alloy workpiece in actual production, the orthogonal grinding experiment of Ti-6Al-4V was established. The milling force in the milling process was measured and recorded using a Kistler three-dimensional dynamometer (model: 9272), and the microstructure of the milling surface was observed using an EBSD detector (model: Nordly-Max-2). Through the analysis of the experimental results, the applicability of the prediction model established in this paper was verified, and the effects of the milling parameters on the milling force and milling surface microstructure were further studied.

## 2. Materials and Methods

### 2.1. Materials and Experimental Equipments

The titanium alloy bars used in this study are obtained by rolling and machined into cuboid samples of 15×15×30mm size by wire cutting. In order to ensure that all the milling samples have the same surface microstructure before machining, all wire cutting samples are obtained from the same Ti-6Al-4V bar. The samples and the experimental equipment used in the milling experiment are shown in Figure 1. The milling force in the milling process is transmitted from the dynamometer to the portable USB signal collector by means of electrical signals. The electrical signals are processed by the charge signal amplifier and then transmitted to the PC.

The components of Ti-6Al-4V (data provided by the supplier) are shown in Table 1. The milling cutter used in the experiment was a YG6X cemented carbide end milling cutter with 4 edges, 10 mm diameter, 9° front angle, 8° back angle, and 35° helix angle. The milling method is climb milling, and the milling process is dry milling (without adding coolant). In order to verify the accuracy of the prediction model, dry milling is used in the experimental design, which is consistent with the mechanical–thermal coupling mode in the prediction model. Before the milling experiment, all milling surfaces are finished with the same face milling cutters.

The material removal rate per unit time is low in the actual milling process of small titanium alloy workpieces. In this study, with reference to the range of processing parameters of Ti-6Al-4V small workpieces, an orthogonal experimental table was designed, as shown in Table 2. In order to investigate the evolution of the micro-texture after milling, EBSD samples of Ti-6Al-4V were prepared after mechanical polishing. In mechanical polishing, the sample surface is polished with 120 μm sandpaper, 3 μm and 0.3 μm polishing cloth until several shallow scratches are formed. The schematic diagram of the EBSD observation is shown in Figure 2.

### 2.2. Prediction Model of the Milling Force

#### 2.2.1. Taylor Factor Model

Schmidt’s law is widely used to study the slip and twin mechanisms of materials. Schmidt’s law holds that there is a critical value of the shear stress in the slip system and twin system, and the slip system and twin system that reach the critical value start first. The Schmidt factor is calculated by the following formula:(1)Schmidt factor=cos(Φ)·cos(Ω)

Here, Φ represents the angle between the slip plane and twin plane and macro-stress, and Ω represents the angle between the slip direction, twin direction, and macro-stress. The greater the Schmidt factor is, the greater the shear stress is, and the material is more likely to slip or twin. According to von Mises law [21], polycrystals usually require at least five independent or systematic systems during plastic deformation. In polycrystal deformation, the complex distribution of the microscopic stress field leads to the anisotropy of the flow stress, and there is a difference between the local stress and macroscopic stress of each grain. The expression of the flow stress is as follows:(2)$$\sigma_{s}=\frac{\tau_{c}}{cos\left(\Phi \right)\cdot cos\left(\Omega \right)}=m_{c}\cdot \tau_{c}$$

Here, $\tau_{c}$ represents the critical shear stress of the activated slip system and twin system, and $m_{c}$ is the reciprocal of the maximum Schmidt factor. For single crystals, the normal stress $\sigma =M\cdot \tau_{c}$, and M is called the Taylor factor [22]. In polycrystals, the Taylor factor changes with the change in the crystal structure and force direction of each grain. Mao et al. [23] and Zhao [24] assumed that the Taylor factor is still applicable in a high-strain-rate machining environment and multi-slip system, and the Taylor factor of a polycrystal is the linear superposition of the single crystal Taylor factor; thus, they proposed the Taylor factor model of the FCC crystal structure and BCC crystal structure. The validity of the Taylor factor model is verified by micro-grinding experiments on 3J33b alloy steel. The single crystal Taylor factor model of the FCC crystal structure and BCC crystal structure is as follows:(3)$$M^{FCC,BCC}=\sum_{i=1}^{N}\frac{1}{cos\left(\Phi_{i}+\gamma_{1}\right)\cdot cos\left(\Omega_{i}+\gamma_{1}^{\prime }\right)}$$

In the above formula, N represents the number of slip systems in the FCC or BCC crystal structure, and $\gamma_{1}$ and $\gamma_{1}^{\prime }$ are correction angles of 0° and 10°, respectively. By regarding the Taylor factor model of the polycrystal as the linear superposition of the single crystal, the Taylor factor model of the polycrystal is obtained as follows:(4)$$M^{Poly}=\sum_{j=1}^{N_{1}}f_{j}\cdot M_{j}^{FCC,BCC}$$

In the formula, $f_{j}$ represents the distribution function value of the grain orientation j, and $N_{1}$ represents the number of the grain orientation.

In this study, on the basis of the above research, the Taylor factor model of the HCP crystal structure for Ti-6Al-4V is established. Ti-6Al-4V is a typical (α+β) titanium alloy, in which α-Ti has an HCP structure, while β-Ti has a BCC crystal structure. Figure 3 shows the crystal structure diagrams of HCP structure (Figure 3a) and BCC structure (Figure 3b).

In the HCP crystal structure, the slip types can be divided into three types: 〈a〉 slip, c slip, and 〈a+c〉 slip. Among them, 〈a〉 slips and 〈a+c〉 slips are easily opened during plastic deformation. Because the 〈a〉 slip can only provide four independent slip systems, which cannot meet the von Mises criterion, the HCP crystal structure also needs to turn on the 〈a+c〉 slip and twin in order to meet the five independent slip system conditions for plastic deformation. The common slip systems of HCP metals are listed in Table 3 [25].

Although the 〈a+c〉 slip can coordinate the deformation in the direction of the c-axis to some extent, the 〈a+c〉 slip is difficult to open and unstable in HCP metal. It is inevitable that the twinning mode must be activated to coordinate the deformation on the c-axis [26,27]. The common twin systems of HCP materials are shown in Table 4 [28]. Many studies have shown that the c/a axial ratio ($\gamma_{c/a}$) has an effect on the selection of deformation twins in HCP materials [29,30,31], and the twins that can be activated in HCP materials with a different $\gamma_{c/a}$ are different. The $\gamma_{c/a}$ of titanium alloy is 1.587, which is less than the ideal axial ratio (1.633). In the five common twinning modes of HCP materials, the $\left\{10\bar{1}2\right\}\langle 10\bar{1}\bar{1}\rangle$ extension twins and $\left\{11\bar{2}2\right\}\langle 11\bar{2}\bar{3}\rangle$ contraction twins mainly occur in titanium alloys.

In the BCC crystal structure, the number of independent slip systems according to von Mises criterion can satisfy the uniform plastic deformation, so that only the role of the slip systems is generally considered. The slip plane families of BCC materials are {101}, and the slip direction families are 〈111〉. All slip systems are shown in Table 5 [32].

Based on the plastic deformation mechanism, the Taylor factor model considering the slip and twin in α-Ti and slip in β-Ti is established in order to describe the effect of the grain orientation of Ti-6Al-4V during plastic deformation. The Taylor factor model of the HCP structure of α-Ti constructed in this study is as follows:(5)$$M^{HCP}=\sum_{i=1}^{N}\frac{1}{cos\left(\Phi_{i}+\gamma_{1}\right)\cdot cos\left(\Omega_{i}+\gamma_{1}^{\prime }\right)}+\sum_{j=1}^{N_{1}}\frac{1}{\cos\left(\Phi_{j}+\alpha_{1}\right)\cdot cos\left(\Omega_{j}+\alpha_{1}^{\prime }\right)}$$

In the formula, *N* represents the number of slip systems, $N_{1}$ represents the number of twin systems, $\gamma_{1}$ and $\gamma_{1}^{\prime }$ are the correction angles, which are 0° and 10°, respectively, [24], and $\alpha_{1}$ and $\alpha_{1}^{\prime }$ are deviation angles of 0° and 3°, respectively [33]. The Taylor factor model of a single crystal with a BCC structure refers to Formula (3). In this study, the Taylor factor model of polycrystals with an HCP structure and a BCC structure is regarded as the linear superposition of the Taylor factors of single crystals, as follows:(6)$$M^{Poly}=\sum_{i=1}^{N_{3}}f_{i}\cdot M_{i}^{HCP,BCC}\;$$

In the formula, $f_{i}$ represents the distribution function value of the grain orientation, and $N_{3}$ refers to the number of the grain orientation. The HCP structure and BCC structure belong to α-Ti and β-Ti, respectively, in Ti-6Al-4V. Thus, the Taylor factor model of the two-phase materials was constructed in this study as follows:(7)$$M=f_{\alpha -Ti}\cdot M^{HCP}+f_{\beta -Ti}\cdot M^{BCC}\;$$

Here, $f_{\alpha -Ti}$ and $f_{\beta -Ti}$ are the proportions of α-Ti and β-Ti in Ti-6Al-4V, respectively.

#### 2.2.2. Milling Force Prediction Model Considering the Taylor Factor

When establishing the analytical model of the milling force, the milling process needs to be simplified using the right angle cutting model or bevel cutting model. On the basis of Oxley’s prediction model, Ji [15] established a cutting force prediction model with a wider range of parameters, considering the thermo-mechanical coupling effects of the material and tool parameters. In this study, the cutting force prediction model considering the Taylor factor was established by combining the cutting force prediction model proposed by Ji [15] with the Johnson–Cook flow stress model, considering the influence of the microstructure.

In the prediction model of the cutting force, the main sources of the cutting force are the chip-forming force and ploughing force. The chip-forming force analytical model and ploughing force model [34] used in this study are shown in Figure 4. According to the analytical model of the chip-forming force, the chip-forming force ($F_{Chip}$) can be divided into the cutting force along the cutting direction ($F_{Cutting}$) and the cutting force perpendicular to the machined surface ($F_{Thrust}$). The components $F_{Cutting}$ and $F_{Thrust}$ can be calculated by the following formula:(8)$$\{\begin{array}{c} F_{Cutting}=R\cdot cos\left(\lambda -\alpha \right)\\ F_{Thrust}=R\cdot sin\left(\lambda -\alpha \right) \end{array}\;$$
(9)$$R=\frac{k_{AB}t_{1}w}{sin\phi \cdot cos\theta }$$

In the formula, R is the resultant force of the normal force and friction force, θ is the angle between R and the shear plane A-B, α is the tool rake angle, ϕ is the shear angle, λ is the angle of friction, $k_{AB}$ is the flow stress on the shear plane A-B, $t_{1}$ is the undeformed chip thickness, and w is the cutting width. Among these, $k_{AB}$ must be calculated by the Johnson–Cook flow stress constitutive equation. The milling process of titanium alloy is a process comprising a large strain, high strain rate, and high temperature, which is consistent with the scope of the application of the Johnson–Cook model. According to the research results of Perez et al. [14], the Zerilli–Armstrong model can also effectively describe the milling process of titanium alloy.

When the material is deformed during processing, the microstructure of the material also evolves, including elements such as the phase distribution, grain size, and grain orientation. The effect of the grain size on the flow stress during machining can be quantified by Hall–Petch formula [35]. According to the Hall–Petch formula, the flow stress is inversely proportional to the square root of the grain size. On the other hand, the Taylor factor model established in this study can quantitatively describe the effects of the phase distribution and grain orientation on the plastic deformation process of materials. Combined with the Hall–Petch formula and the influence model of the microstructure on the flow stress [36,37,38], the Johnson–Cook flow stress model, considering the initial microstructure of the material, is used to calculate the flow stress in the cutting process as follows:(10)$$\sigma =\left(A+B\varepsilon_{p}^{n}\right)\left(1+Cln\frac{\dot{\varepsilon_{p}}}{\dot{\varepsilon_{0}}}\right)\left(1-{\left(\frac{T-T_{w}}{T_{m}-T_{w}}\right)}^{m}\;\right)+M\alpha Gb\sqrt{\rho_{1}}+\frac{K_{HP}}{\sqrt{D_{d}}}\;$$
(11)$$K_{HP}=M\cdot \sqrt{\frac{\tau_{b}\cdot 4G\cdot b}{\left(1-\nu \right)\pi }}$$

In the formula, *A*, *B*, *C*, *n*, and *m* are all material constants in the Johnson–Cook model, $\varepsilon_{p}$ is the effective plastic strain, $\dot{\varepsilon_{p}}$ is the effective strain rate, and $\dot{\varepsilon_{0}}$ is the reference strain rate (usually 1$s^{-1}$). *T* is the current temperature, $T_{w}$ is the ambient temperature, and $T_{m}$ is the material melting point. *M* is the Taylor factor model, *G* is the material shear modulus, *b* is the Burgers vector, $\rho_{1}$ is the dislocation density, ν is Poisson’s ratio, $\tau_{b}=0.057G$, α is the dislocation density interaction coefficient (α=2(1−υ)/(2−υ)) [37], and $D_{d}$ is the diameter of the grain. According to the Johnson–Cook flow stress model, the flow stress of the shear plane AB ($k_{AB}$) can be expressed as:(12)$$k_{AB}=\frac{\sigma_{AB}}{\sqrt{3}}\;$$

In this study, the analytical model of the ploughing force proposed by Waldorf et al. [34] was used to predict the ploughing force caused by the fillet of the knife tip. The ploughing force can also be divided into two components, including the ploughing force $P_{Cutting}$ along the cutting direction and the ploughing force $P_{Thrust}$ perpendicular to the machined surface, as follows:(13)$$P_{Cutting}=k_{AB}\cdot w\cdot [cos\left(2\eta_{plow}\right)cos\left(\phi -\gamma_{plow}+\eta_{plow}\right)+\;\left(1+2\theta_{fan}+2\gamma_{plow}+sin\left(2\eta_{plow}\right)\right)\cdot sin\left(\phi -\gamma_{plow}+\eta_{plow}\right)]\cdot CA\;$$
(14)$$P_{Thrust}=k_{AB}\cdot w\cdot [\left(1+2\theta_{fan}+2\gamma_{plow}+sin\left(2\eta_{plow}\right)\right)\;\cdot cos\left(\phi -\gamma_{plow}+\eta_{plow}\right)-cos\left(2\eta_{plow}\right)sin\left(\phi -\gamma_{plow}+\eta_{plow}\right)]\cdot CA$$
(15)$$CA=\frac{R_{fan}}{sin\eta_{plow}}$$

The fan-shaped angles $\theta_{fan}$, $\gamma_{plow}$, $\rho_{prow}$, and $\eta_{plow}$ can be solved according to the geometric relation and friction relation [34]. Here, *r* is the radius of the tool edge and ϕ is the shear angle. $R_{fan}$ is solved by the following equation:(16)$$R_{fan}=\sqrt{{\left(r\cdot tan\left(\frac{\pi +2\alpha }{4}\right)+\frac{\sqrt{2}\cdot R_{fan}\cdot sin\left(\rho_{prow}\right)}{tan\left(\frac{\pi +2\alpha }{2}\right)}\right)}^{2}+2{\left[R_{fan}\cdot sin\left(\rho_{prow}\right)\right]}^{2}}\cdot sin\left(\eta_{plow}\right)\;$$

After the calculation of the chip-forming force and ploughing force is completed, the total cutting force in the cutting process can be expressed as:(17)$$\{\begin{array}{c} F_{t}=F_{Cutting}+P_{Cutting}\\ F_{n}=F_{Thrust}+P_{Thrust} \end{array}\;$$

## 3. Results and Discussion

### 3.1. Analysis of the Milling Force and Verification of the Prediction Model

The physical parameters of the workpiece material Ti-6Al-4V are shown in Table 6. The Johnson–Cook flow stress model, considering the microstructure of materials, is required in the cutting force prediction model. The Johnson–Cook material parameters of Ti-6Al-4V are shown in Table 7 [39]. The material parameters of the tool are shown in Table 8, below.

Microstructure maps of the material surface before milling are shown in Figure 5. From the figure, we can observe that the grains on the material surface are mainly equiaxed grains, the average grain size is about 4.5μm, the proportion of the α phase is about 98.6%, and the proportion of the β phase is about 1.4%. Figure 5d shows the KAM map of the surface. According to the KAM diagram, the degree of deformation can be analyzed intuitively. The higher the KAM value is, the greater the degree of plastic deformation is.

In this study, the geometric dislocation density ($\rho^{GND}$) was calculated according to the KAM value, which is expressed by the following formula:(18)$$\{\begin{array}{c} \rho^{GND}=\frac{2KAM_{avg}}{\mu b}\\ KAM_{avg}=\exp\left(\frac{1}{N}\overset{i}{\underset{1}{\sum }}lnKAM_{i}\right) \end{array}\;$$

In the formula, μ is the scanning step, b is the Burgers vector, $KAM_{avg}$ is the average KAM value of the selected area, $KAM_{i}$ represents the KAM value at point i, and N represents the number of points in the test area.

In this study, the milling force was collected by the Kistler-9272 three-dimensional dynamometer, which can collect and record F_X_, F_Y_, F_Z_, and the corresponding torque. In order to eliminate the influence of the noise signal caused by the spindle vibration, the collected data must be transformed by FFT (fast Fourier transform). Figure 6 shows the signal curve of the normal milling force in the first group of experiments before and after low frequency filtering, with a fixed frequency of 42 Hz.

The experimental parameters and results under the condition of dry milling are shown in Table 9.

In order to intuitively analyze the effect of each milling parameter on the actual milling force $F_{t}$ and $F_{n}$, the range analysis of the milling force was carried out in this study. In the analysis, the K value of each level represents the sum of the experimental results of the selected factors at the current level. The K-avg value indicates the average value of K corresponding to the selected factor at the current level, and the R value represents the range of each factor, which is obtained by subtracting the minimum value from the maximum value in the K-avg. The higher the R value is, the more obvious the influence of this factor on the result is. The range analysis tables of the milling force measured in the experiment are shown in Table 10 and Table 11, respectively. By analyzing the results of the orthogonal experiment, we can see that, for the tangential component of the cutting force $F_{t}$, the ranges of the linear speed, feed rate, and cutting depth are 21.90, 9.73, and 22.20, respectively. The ranges of the linear velocity, feed rate, and cutting depth of the normal component of the cutting force $F_{n}$ are 3.10, 5.40, and 11.13, respectively. Therefore, the influence of the cutting depth on $F_{t}$ is the greatest, the influence of the linear velocity is the second greatest, and the influence of the feed rate is the least. For $F_{n}$, the influence of the cutting depth is the greatest, the influence of the feed rate is the second greatest, and the influence of the linear velocity is the least. The above analysis shows that the milling depth has the greatest influence on the overall milling force in the process of milling.

In the actual milling process, because the surface of the machined material is not completely horizontal, there is a difference between the actual machining depth and the theoretical value of each sample in the milling process. Moreover, the tool wears during the machining process, which changes the geometric parameters of the tool (rake angle, edge radius, etc.) and then affects the milling force. The milling force prediction model used in this study is based on the assumption that the milling parameters (cutting speed, feed rate, cutting depth) and tool geometric parameters do not change in the machining process; thus, there is an error between the milling force prediction model and the actual experimental results. Generally, when the actual milling depth is greater than the parameters set in the model, or when the tool is worn, the actual milling force will be greater than the predicted value of the model. When the actual milling depth is less than the parameters in the model, or a new tool is replaced, the actual milling force will be less than the predicted value of the model.

Figure 7 shows that, within the given range of the milling parameters, both the tangential milling force and the normal milling force show an overall increasing trend with the increase in the milling depth and feed rate. The tangential milling force decreases with the increase in the milling speed, and the normal milling force first decreases and then increases slightly with the increase in the milling speed. In the cutting direction, both the chip-forming force and the ploughing force increase with the increase in the feed rate. This is due to the increase in the feed, which is the thickness of the cut, and, therefore, the increase in the cutting force. However, when the feed rate increases, the deformation coefficient of the chip decreases, and the friction coefficient also decreases, thereby reducing the friction force. Due to these effects, the increasing speed of the cutting force is not proportional to the increasing speed of the feed. When the cutting speed increases, the cutting force in the cutting direction decreases. As the cutting speed increases, the cutting temperature also increases, and the strength and hardness of the processed metal decrease, resulting in a decrease in the cutting force. The cutting speed and the feed amount have almost no effect on the cutting force in the direction perpendicular to the surface of the workpiece.

Many researchers have found that the milling force decreases with the increase in the milling speed in their experiments. The research results of Klocke et al. [40] indicate that, on the one hand, intermittent cutting increases with the increase in the milling speed, and the hardening of the material will lead to the increase in the deformation resistance in the area affected by the shear force. On the other hand, the increase in the milling speed leads to the increase in the milling temperature and the softening of the materials, which leads to the decrease in the milling force. The influence of the milling speed on the milling force is the result of the comprehensive action of the above two factors. The increase in the feed rate and milling depth affect the geometry of the milling area, resulting in the increase in the cross-sectional area of the milling area and the amount of material removal per unit time, which leads to the change in the milling force. Many scholars have studied the milling force of Ti-6Al-4V, and most of the analytical results concerning the influence of the milling parameters on the milling force correspond to the results of this study [41,42].

Figure 8 shows a comparison of all the experimental results with those predicted by the model. According to the analysis of Figure 8, it is found that there is a small error between the predicted and experimental values of the cutting force in both the tangential and normal directions. The reason may be that the change in the milling force caused by tool wear and the change in the milling depth caused by the horizontal degree of the machined surface are not considered in theoretical research.

Table 12 shows the relative errors of the milling forces and predictions for each group. From the table, we can see that the relative error range of the tangential milling force obtained from the prediction model in this study is 3.4–20.7%, and the average error is 8.3%. And the relative error range of the normal milling force is 3.7–14.3%, and the average error is 8.5%.

From Table 12, we can see that when the milling speed and feed rate are at a high level but the milling depth is small, the relative error between the predicted results and the experimental results increases (14.2%, 20.7%, etc.). This may be caused by the following factors: (1) Firstly, when the milling depth decreases, the influence of the chip-forming force on the milling force decreases, which leads to the greater influence of the ploughing force on the milling force. The ploughing force is also affected by the material hardness, but the material hardness is not considered in the prediction model established in this study, which may lead to the increase in the relative error. (2) Secondly, when the milling depth is very low, the change in the actual milling depth caused by the uneven surface of the material has a great influence on the results. The prediction model cannot simulate the change in the milling depth in the machining process, which also leads to the increase in the relative error.

The average prediction errors obtained by Ji [15] and Li et al. [43] through the cutting force prediction model, without considering the Taylor factor, are 13.4% and 10.2%, respectively. This indicates that the prediction accuracy of the milling force prediction model considering the Taylor factor proposed in this study is better than the original model.

### 3.2. Effects of the Milling Parameters on the Microstructure Evolution of the Milling Surface

#### 3.2.1. Microstructure Shape and Grain Size

In the milling process, the change in the milling parameters lead to changes in the temperature, strain, and strain rate of the machined surface, thus affecting the grain structure of the machined surface. Under the action of milling, some of the grains in the cutting are distorted along the milling direction of the tool. The grain orientation distribution maps and grain size statistics of Ti-6Al-4V under different milling parameters are shown in Figure 9 and Figure 10 (subfigures 1–9 correspond to 1–9 groups of samples in the orthogonal experiment table, respectively). The research object is the milling surface of the sample, and the EBSD calibration area is 60 μm × 50 μm, which characterizes the local microstructure of the milling surface of the sample. From Figure 9, we can see that the grains in the surface of the milled materials are refined, and most of them are equiaxed grains.

By analyzing the statistical diagram of the grain size, it is found that the grain size decreases with the increase in the milling speed and milling depth, and the effect of the milling depth on the grain size is the most significant. Further analysis shows that the grain size of the subsurface layer is smaller than that of the surface layer, and the grain size of the machined surface shows a gradient change in the depth direction. The change in the grain size is closely related to the temperature field and strain rate field distribution of the material surface. On the machined surface, which is in direct contact with the tool, the plastic deformation caused by the friction and shear between the tool and the material and the high temperature caused by milling has obvious effects on the grain size. Due to the high temperature produced by the machined surface during milling, the refined grains become larger under the action of the high temperature. Meanwhile, most of the heat generated in the milling process is carried away by the chip, and in the subsurface layer of the material surface, the high strain rate plays a dominant role in shaping the grain size. When the milling speed increases from 60 m/min to 120 m/min, the milling temperature of the material does not change obviously in this range of the milling speed because the contact time between the tool and the material becomes shorter and more chips reduce the heat generated in the milling. Therefore, in this study, with the increase in the milling speed, the effect of plastic deformation on the grain size plays a major role, and the grain size decreases with the increase in the milling speed. Similarly, with the increase in the milling depth, the plastic deformation increases significantly, and the strain rate field of the machined surface layer is transferred to the subsurface layer, causing grain refinement.

The above results show that the change in the grain size follows the law of grain refinement on the micron scale, as follows: with the enhancement of the plastic deformation, the grain is stretched, distorted, and broken, and the grain refinement is enhanced, while with the increase in the temperature, the grain easily recrystallizes and grows, and the grain refinement is weakened. Moreover, there are no twins in most grains and flat twins in a few grains, indicating that the deformation twinning behavior occurs in the grains to coordinate the plastic deformation. In the milling process of Ti-6Al-4V, dislocation slip plays a dominant role in the plastic deformation of the surface, while the deformation twins coordinate only a small part of the plastic deformation process, and the grains of almost all the samples are refined to varying degrees, which can be considered as the result of the complex interaction between the twins and dislocations.

#### 3.2.2. Grain Boundary Distribution and Phase Distribution

The grain boundary distribution maps and phase distribution maps of Ti-6Al-4V under different milling parameters are shown in Figure 11 and Figure 12 (subfigures 1–9 correspond to 1–9 groups of samples in the orthogonal experiment table, respectively). According to the misorientation angle difference of the adjacent grains, the grain boundary is divided into the high-angle grain boundary (misorientation angle ≥15°, HAGB) and low-angle grain boundary (misorientation angle <15°, LAGB). In Figure 11, the HAGB is represented by blue lines, the LAGB is represented by green lines, and the grain boundaries of the sub-grains are represented by red lines. Based on the analysis of the proportion of each grain boundary, it was found that the proportion of the LAGB increases with the increase in the milling depth. With the increase in the milling speed and feed rate, the proportion of LAGB first increases and then decreases. Further analysis showed that the influence of the milling depth on the LAGB distribution is the greatest.

Grain refinement caused by plastic deformation is characterized by the appearance of a large LAGB within the grains, while the grains undergoing dynamic recrystallization usually have an HAGB. The percentage of the LAGB can reflect the intensity of the dislocation activity. When the milling speed, feed rate, and milling depth increase, the amount of strain and the strain rate increase, and the percentage of the LAGB increases. This indicates that the proportion of refined grains caused by plastic deformation gradually increases, the slip system inside the grains is activated, and the dislocation slip is the main deformation mode. With further increases in the milling speed and feed, the percentage of the LAGB tends to decrease, which may be due to the shortened time given for dislocations to occur inside the grains due to the increase in the strain rate. The time given for dislocation slip to occur inside the grain is reduced, and the grain does not have sufficient time to open the slip system so as to coordinate the deformation. Moreover, with the increase in the strain amount and strain rate, the heat generated by the milling surface also increases, and the grains are more likely to undergo dynamic recrystallization under the action of the high temperature, resulting in an increase in the percentage of the HAGB. Some studies have also pointed out that with the increase in the degree of grain deformation, a large number of dislocations in the grain will offset each other and rearrange, which also reduces the percentage of the LAGB to a certain extent.

As mentioned above, Ti-6Al-4V is a two-phase (α+β) titanium alloy. The phase distributions of each group of the sample milling surface are shown in Figure 12, with the α phase in red and the β phase in green. The proportion of the α phase in the milling surface of each group is higher than 90%. Compared with the phase distribution of the surface of the material before milling, this indicates that the proportion of the α phase and β phase changed to varying degrees under the action of milling. With the increase in the milling speed and milling depth, the proportion of β phase increases, and the milling depth has the most significant effect on the beta phase distribution.

The cutting temperature and local stress are the main factors causing Ti-6Al-4V phase transformation. In the milling process, the material passes through two stages; the first stage is the heat generation stage, caused by instantaneous pressure, and the second stage is the cooling stage. In the first stage, there is a transition from the α phase to the β phase, while in the second stage, there is a transition from the β phase to the α phase. The milling method used in this study was dry milling, without the effect of a coolant, and the effect of the cooling stage is not obvious. The above analysis only considers the effect of the temperature on the phase transformation, but in the actual milling process, the severe plastic deformation and the mechanical load imposed on the material by the cutting tool also transform the α phase into the β phase. Due to the fact that the α-Ti has an HCP structure and the β-Ti has a BCC structure, the number of independent slip systems of α-Ti is much less than that of β-Ti. In other words, β-Ti is more prone to plastic deformation than α-Ti under the same milling force, which can also explain the increased proportion of the β phase in the milling process.

#### 3.2.3. Texture Evolution and Twinning Behavior

Figure 13 (subfigures 1–9 correspond to 1–9 groups of samples in the orthogonal experiment table, respectively) shows the pole figures and twin grain boundary contrast maps of the Ti-6Al-4V surface under different milling parameters. The orientation distribution of the α−Ti grains perpendicular to the ND direction under different milling conditions is represented by the {0001} pole figure, $\left\{10\bar{1}0\right\}$ pole figure, and {1120} pole figure. In the IPF (Figure 9), most of the areas are blue, indicating that the grain orientation distribution of the $\left\{10\bar{1}0\right\}$ crystallographic planes in the vertical ND direction accounts for a large proportion. From the corresponding pole figure ($\left\{10\bar{1}0\right\}$ pole figure), the maximum intensity of most textures appears in the ND direction, indicating that there is a large number of prismatic textures on this cross section. Further analysis indicated that with the increase in the milling depth, the intensity of the prismatic surface texture increases (in Figure 9(3),(4),(7)). Moreover, with the increase in the milling depth, the texture of the $\left\{10\bar{1}0\right\}$ crystallographic plane aggregates to the [0001] orientation, and the main texture distribution is in the range of ±45° inclination from the [0001] orientation towards the TD and RD direction.

According to the analysis of the {0001} pole figure, there is no obvious basal texture of the milling surface, and the main component is the [0001] orientation, with an inclination of ±45°~60° towards TD and RD. When the feed rate reaches 240mm/min, a small degree of low-intensity basal texture can be observed in the {0001} pole figure (in Figure 9(6), Figure 9(9)). The results show that the texture intensity of the {0001} pole figure decreases with the increase in the milling speed, and with the increase in the milling depth, the texture orientation of the {0001} crystallographic plane is inclined towards the RD and TD direction (in Figure 9(3),(4),(7)).

In addition, some textures with a high intensity are also observed in the {1120} pole figure—that is, the [0001] orientation with a tilt of ±20°~30° towards TD and RD. With the change in the milling parameters, the texture distribution of the {1120} plane shows no obvious regularity, but the texture intensity of the {1120} plane decreases with the increase in the milling speed. The texture intensity also increases slightly with the increase in the milling depth (in Figure 9(3),(4)). Compared with the texture intensity before milling (Figure 5), it was found that the texture intensity changes in varying degrees with the change in the milling speed and milling depth. Among all the milling parameters, the effects of the milling speed on texture strength are the most significant, and the overall intensity of the texture decreases with the increase in the milling speed.

In Figure 14, the red line, green line, yellow line, and blue line represent the twin boundaries of $\left\{10\bar{1}2\right\}$, $\left\{11\bar{2}1\right\}$, $\left\{11\bar{2}2\right\},$ and $\left\{10\bar{1}1\right\}$, respectively. By observing the selected region, it is found that the content of all the twin grain boundaries is very low, and the volume fraction amounts to less than 1%, so that only two twin grain boundary distribution maps were selected to be illustrated. In all twin modes, the main activated twins are $\left\{10\bar{1}1\right\}$ and $\left\{10\bar{1}2\right\}$, and the number of $\left\{10\bar{1}1\right\}$ twins is much higher than that of $\left\{10\bar{1}2\right\}$, while a small number of $\left\{11\bar{2}1\right\}$ twins can also be observed in Figure 14b. On the whole, when milling under the milling parameters selected in this study, the strain in the grain is coordinated mainly by the dislocation slip, and the contribution of the twins is very minimal.

## 4. Conclusions

1. The influence of the cutting depth on $F_{t}$ is the greatest, the influence of the linear velocity is the second greatest, and the influence of the feed rate is the least. For $F_{n}$, the influence of the cutting depth is the greatest, the influence of the feed rate is the second greatest, and the influence of the linear velocity is the least. In the process of milling, the milling depth has the greatest influence on the overall grinding force.

2. Within the given range of the milling parameters, both the tangential milling force and the normal milling force show an overall increasing trend with the increase in the milling depth and feed rate. The tangential milling force decreases with the increase in the milling speed, and the normal milling force decreases first and then increases slightly with the increase in the milling speed.

3. The average error of the tangential milling force obtained from the prediction model in this study is 8.3%, and the average error of the normal milling force is 8.5%. This indicates that the prediction accuracy of the milling force prediction model considering the Taylor factor proposed in this study is better than the original model. The error may be caused by the tool wear and the change in the milling depth (caused by the horizontal degree of the machined surface), which are not considered in theoretical research.

4. The grain size decreases with the increase in the milling speed and milling depth, and the effect of the milling depth on grain size is the most significant. The proportion of the LAGB increases with the increase in the milling depth. Further analysis indicated that the influence of the milling depth on the LAGB distribution is the greatest. With the increase in the milling speed and milling depth, the proportion of the β phase increases, but with the increase in the feed rate, the proportion of the β phase decreases. Moreover, the milling depth has the most significant effect on the beta phase distribution. The texture intensity decreases with the increase in the milling speed and increases with the increase in the feed rate. Among all the milling parameters, the effect of the milling speed on the texture strength is the most significant. When milling under the milling parameters selected in this study, the strain in the grain is coordinated mainly by the dislocation slip, and the contribution of the twins is very minimal.

## Figures and Tables

**Figure 1 micromachines-13-01618-f001:**
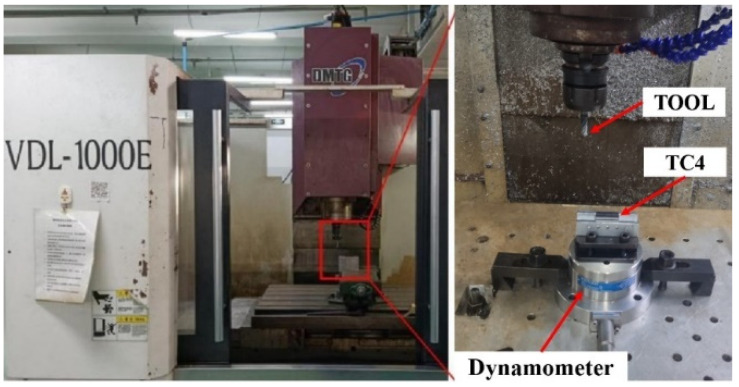
The samples and the experimental equipment used in the milling experiment.

**Figure 2 micromachines-13-01618-f002:**
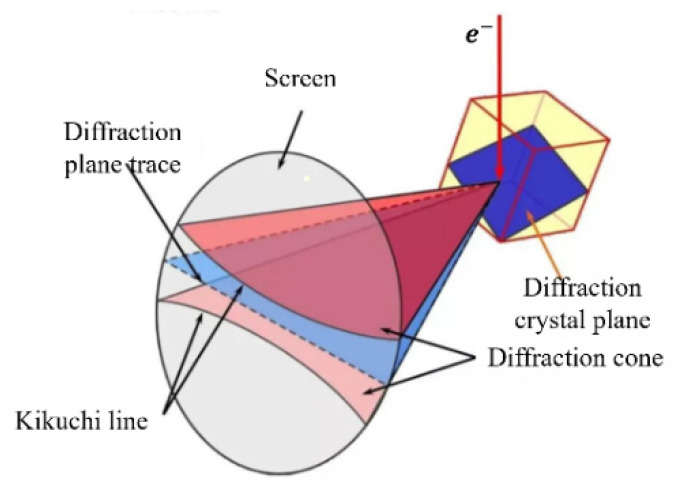
Schematic representation of the EBSD of single crystal planes.

**Figure 3 micromachines-13-01618-f003:**
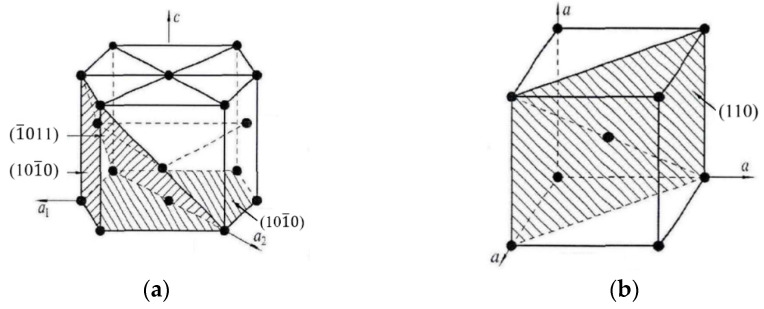
Schematic diagram of the crystal structures of α-Ti and β-Ti. (**a**) α-Ti. (**b**) β-Ti.

**Figure 4 micromachines-13-01618-f004:**
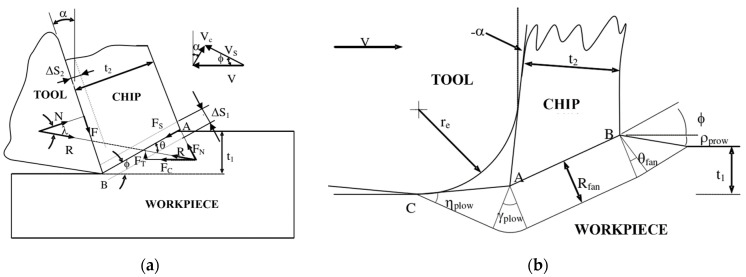
Geometry of the milling force model. (**a**) Chip-forming force model [15]. (**b**) Ploughing force model [34].

**Figure 5 micromachines-13-01618-f005:**
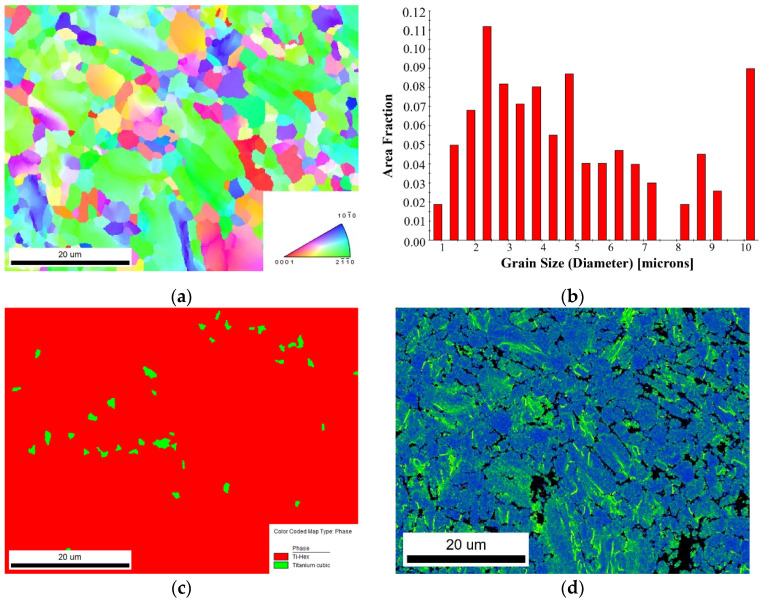
Microstructure maps of the Ti-6Al-4V surface before milling. (**a**) The grain orientation distribution map. (**b**) The grain size statistic map. (**c**) The phase distribution map. (**d**) The KAM map.

**Figure 6 micromachines-13-01618-f006:**
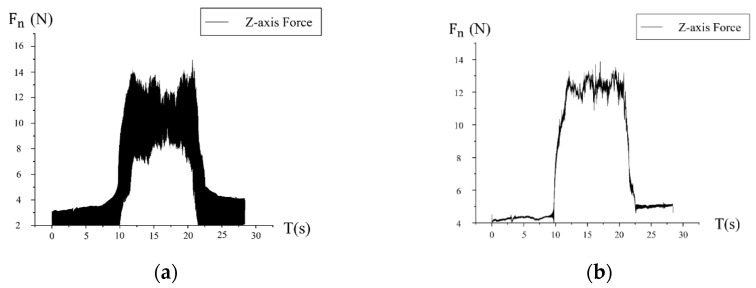
Low frequency filtering of the milling force signal. (**a**) Before FFT. (**b**) After FFT.

**Figure 7 micromachines-13-01618-f007:**
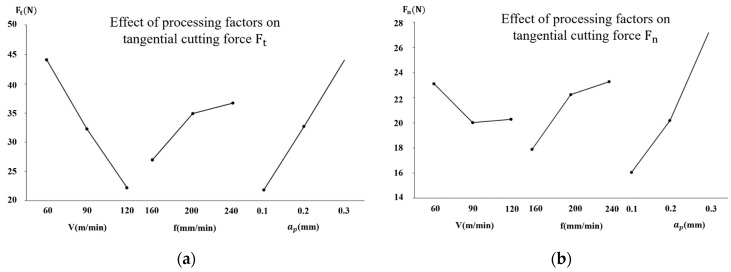
Effect of the level of the milling parameters on the milling force. (**a**) Tangential milling force $F_{t}$. (**b**) Normal milling force $F_{n}$.

**Figure 8 micromachines-13-01618-f008:**
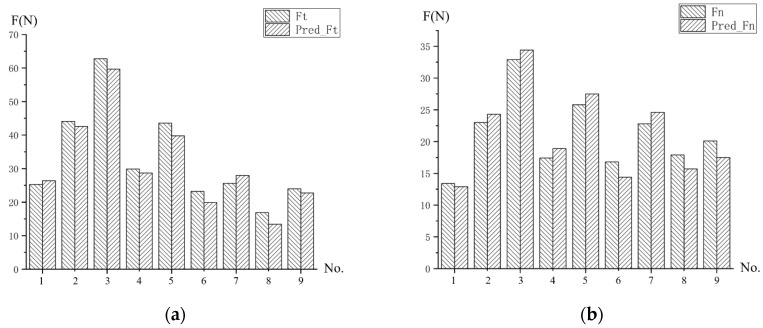
Comparison between the experimental results and predicted values. (**a**) Tangential milling force $F_{t}$. (**b**) Normal milling force $F_{n}$.

**Figure 9 micromachines-13-01618-f009:**
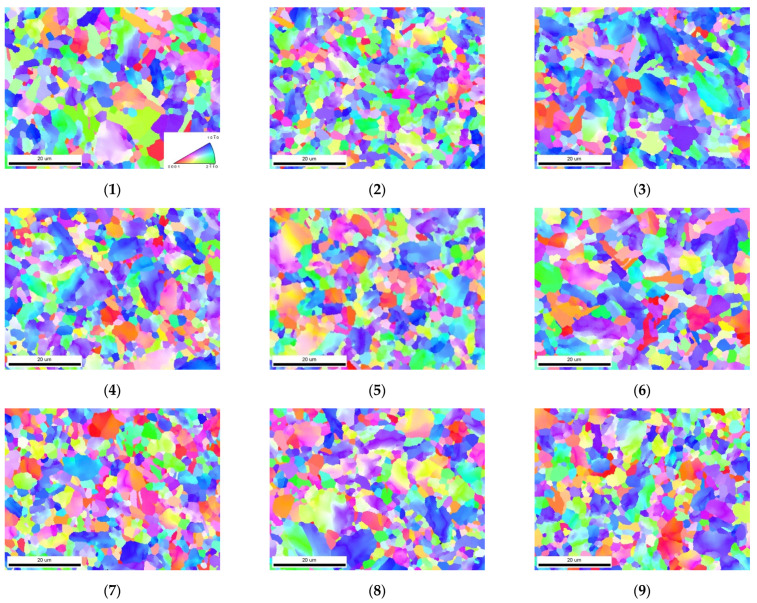
The grain orientation distribution maps of Ti-6Al-4V under different milling parameters.

**Figure 10 micromachines-13-01618-f010:**
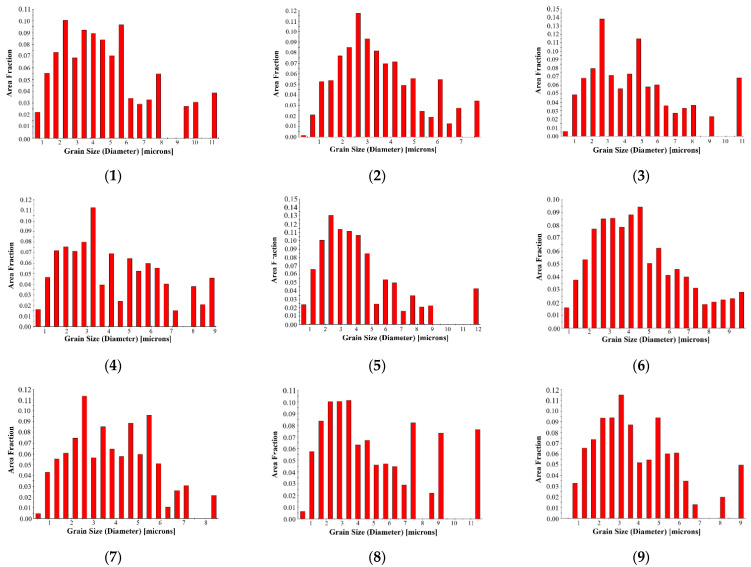
The grain size statistics of Ti-6Al-4V under different milling parameters.

**Figure 11 micromachines-13-01618-f011:**
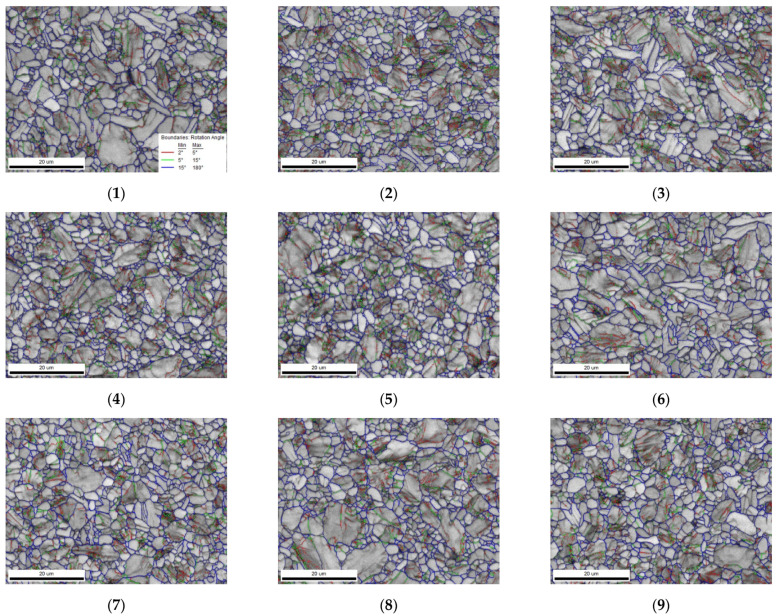
The grain boundary distribution maps of Ti-6Al-4V under different milling parameters.

**Figure 12 micromachines-13-01618-f012:**
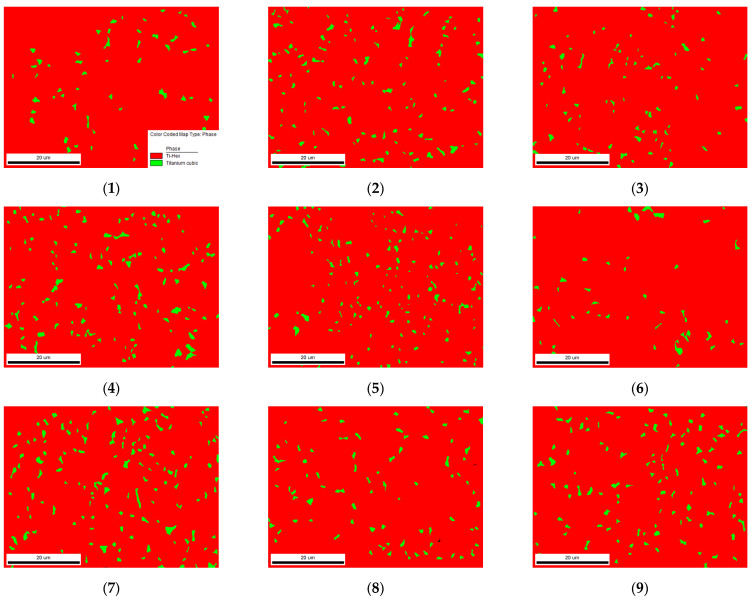
The grain boundary distribution maps of Ti-6Al-4V under different milling parameters.

**Figure 13 micromachines-13-01618-f013:**
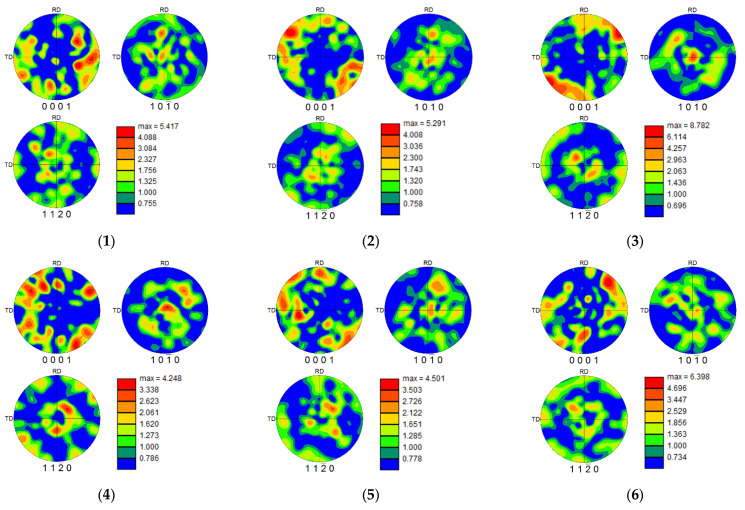
Pole figures of the Ti-6Al-4V surface under different milling parameters.

**Figure 14 micromachines-13-01618-f014:**
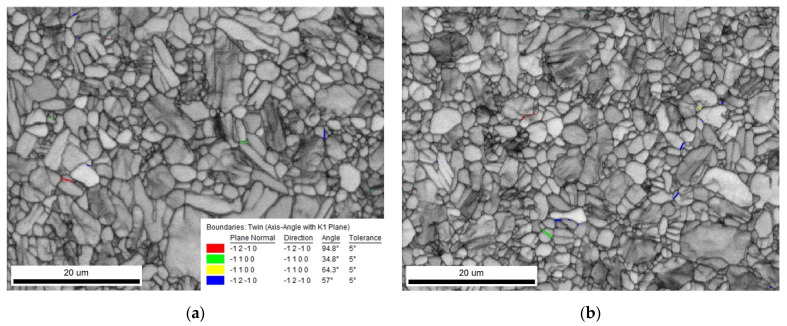
Pole figures of the Ti-6Al-4V surface under different milling parameters. (**a**) Sample 1. (**b**) Sample 9.

**Table 1 micromachines-13-01618-t001:** The main chemical composition of Ti-6Al-4V.

	Al	V	Fe	Si	C	N	H	O	Ti
**wt%**	6.13	3.93	0.103	0.11	0.007	0.035	0.013	0.12	Bal.

**Table 2 micromachines-13-01618-t002:** Orthogonal experimental parameters.

	Factors	$V_{c}\;\left(m/min\right)$	f (mm/min)	$a_{p}\;\left(mm\right)$
Levels				
1		60	160	0.1
2		90	200	0.2
3		120	240	0.3

**Table 3 micromachines-13-01618-t003:** Main slip systems and corresponding parameters of α-Ti [25].

Direction	Types	Crystallographic Elements	Independent Slip System
〈a〉	Basal slip	$\left(0002\right)\langle 11\bar{2}0\rangle$	2
〈a〉	Prismatic slip	$\left\{10\bar{1}0\right\}\langle 11\bar{2}0\rangle$	2
〈a〉	Pyramidal slip	$\left\{10\bar{1}1\right\}\langle 11\bar{2}0\rangle$	4
〈a+c〉	Pyramidal slip	$\left\{\mathrm{hkil}\right\}\langle 11\bar{2}\bar{3}\rangle$	5

**Table 4 micromachines-13-01618-t004:** Main twinning systems and corresponding parameters of α-Ti [28].

Types	Twin Modes	Rotation Angle (°)	Rotation Axis
Extension twin	$\left\{10\bar{1}2\right\}\langle 10\bar{1}\bar{1}\rangle$	84.78	$\langle 11\bar{2}0\rangle$
	$\left\{11\bar{2}\bar{1}\right\}\langle 11\bar{2}\bar{6}\rangle$	35.10	$\langle 10\bar{1}0\rangle$
Contraction twin	$\left\{10\bar{1}1\right\}\langle 10\bar{2}0\rangle$	57.22	$\langle 11\bar{2}0\rangle$
	$\left\{11\bar{2}2\right\}\langle 11\bar{2}\bar{3}\rangle$	64.62	$\langle 10\bar{1}0\rangle$
	$\left\{11\bar{2}4\right\}\langle 22\bar{4}\bar{3}\rangle$	76.66	$\langle 10\bar{1}0\rangle$

**Table 5 micromachines-13-01618-t005:** Main slip systems and corresponding parameters of β -Ti [32].

Slip Plane	Slip Direction
(101)	$\left[\bar{1}\bar{1}1\right]\left[\bar{1}11\right]$
$\left(10\bar{1}\right)$	$\left[1\bar{1}1\right]\left[111\right]$
(011)	$\left[\bar{1}\bar{1}1\right]\left[1\bar{1}1\right]$
$\left(0\bar{1}1\right)$	$\left[\bar{1}11\right]\left[111\right]$
$\left(1\bar{1}0\right)$	$\left[\bar{1}\bar{1}1\right]\left[111\right]$
(110)	$\left[1\bar{1}1\right]\left[\bar{1}11\right]$

**Table 6 micromachines-13-01618-t006:** The physical parameters of Ti-6Al-4V.

Materials	$\rho \;\left(kg/m^{3}\right)$	E (Gpa)	υ	λ (W/m·K)	C (J/kg·K)
Ti-6Al-4V	4430	109 (50 ℃)	0.34	6.8 (20 ℃)	611 (20 ℃)

**Table 7 micromachines-13-01618-t007:** The Johnson–Cook parameters of Ti-6Al-4V [39].

Materials	A (Mpa)	B (Mpa)	C	n	m	$T_{m}\;\left(℃\right)$	$T_{w}\;\left(℃\right)$	α	b(m)	$G\;\left(N/m^{2}\right)$
Ti-6Al-4V	418	394	0.035	0.47	1	1650	25	0.8	$2.86\times 10^{-10}$	$2.05\times 10^{10}$

**Table 8 micromachines-13-01618-t008:** The material parameters of the tool.

Materials	$\rho \;\left(kg/m^{3}\right)$	E (Mpa)	υ	λ (W/m·K)	C (J/kg·K)
YG6X	14,600	640,000	0.22	79.6	176

**Table 9 micromachines-13-01618-t009:** Experimental results of Ti-6Al-4V under dry milling.

No.	$V_{c}$ (m/min)	f (mm/min)	$a_{p}$ (mm)	$F_{t}$ (N)	$F_{n}$ (N)	$Pred\;F_{t}$ (N)	$\mathrm{Pred}\;F_{n}$ (N)
1	60	160	0.1	25.3	13.4	26.4	12.9
2	60	200	0.2	44.1	23.0	42.6	24.3
3	60	240	0.3	62.8	32.9	59.7	34.4
4	90	160	0.2	29.9	17.4	28.7	18.9
5	90	200	0.3	43.6	25.8	39.8	27.5
6	90	240	0.1	23.2	16.8	19.9	14.4
7	120	160	0.3	25.6	22.8	28.0	24.6
8	120	200	0.1	16.9	17.9	13.4	15.7
9	120	240	0.2	24.0	20.1	22.7	17.5

**Table 10 micromachines-13-01618-t010:** Range analysis of the tangential milling force $F_{t}$.

Item	Level	$V_{c}\;\left(m/min\right)$	f (mm/min)	$a_{p}\;(\mathrm{mm})$
K	1	132.2	80.8	65.4
	2	96.7	104.6	98
	3	66.5	110	132
K-avg	1	44.07	26.93	21.80
	2	32.23	34.87	32.67
	3	22.17	36.67	44.00
R		21.90	9.73	22.20

**Table 11 micromachines-13-01618-t011:** Range analysis of the normal milling force $F_{n}$.

Item	Level	$V_{c}\;\left(m/min\right)$	f (mm/min)	$a_{p}(\mathrm{mm})$
K	1	69.3	53.6	48.1
	2	60.0	66.7	60.5
	3	60.8	69.8	81.5
K-avg	1	23.10	17.87	16.03
	2	20.00	22.23	20.17
	3	20.27	23.27	27.17
R		3.10	5.40	11.13

**Table 12 micromachines-13-01618-t012:** The relative errors of the milling forces and predictions for each group.

No.	$F_{t}$ (N)	$Pred\;F_{t}$ (N)	Relative Error (%)	$F_{n}$ (N)	$\mathrm{Pred}\;F_{n}$ (N)	Relative Error (%)
1	25.3	26.4	4.3	13.4	12.9	3.7
2	44.1	42.6	3.4	23.0	24.3	5.7
3	62.8	59.7	4.9	32.9	34.4	4.6
4	29.9	28.7	4.0	17.4	18.9	8.6
5	43.6	39.8	8.7	25.8	27.5	6.6
6	23.2	19.9	14.2	16.8	14.4	14.3
7	25.6	28.0	9.4	22.8	24.6	7.9
8	16.9	13.4	20.7	17.9	15.7	12.3
9	24.0	22.7	5.4	20.1	17.5	12.9

## Data Availability

The author declares that all data are available.

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
