# Peer review of "A Ti-6Al-4V Milling Force Prediction Model Based on the Taylor Factor Model and Microstructure Evolution of the Milling Surface"

_micromachines, 2022, doi:10.3390/mi13101618_

Round 1
Reviewer 1 Report
· Honestly, this is not a bad work but the main one is messed, Perez published the real effect of anisotropy and grain orientation in the dual approach: https://doi.org/10.1016/j.ijmachtools.2021.103801 and the correlated Urbikain works about all type of barrel, ball-end and end-milling tools.
· Pole diagrams are key for that, but the discussion above is better, in forged or cast material, similar performance and behaviors is expected.
· TC4 is better known as Ti6Al4v as it was explained in many works by O.Pereira also missed, about machining with cryogenics. Please change the title. Nobody use dry milling with titanium, please discuss it with respect high pressure coolant and cryogenics with CO2.
· Figure 1 is poor, what are the boxes?
Predicted values using J&C are orientate ones, many discussion is about accuracy. Experimental values are much better.
Cutting speed symbol is vc…..check all the symbols please. I recommend ISO standards or Lamikiz, Urbikain papers to follow a regular and ISo names’
Cutting tools has helix angle…so why only three forces: define in detail the cutting tool.
Your first conclusion must be discussed: feed and depth of cut form the chip section.
inear velocity. In classic views it does not have influence.
My opinion: aims at the pole and structure ideas,
The average error of tangential milling force obtained from the prediction model 413 of this study is 8.3%, and the average error of normal milling force is 8.5%
State of the art must be enriched, with above works and many other about titanium machining.
Author Response
The author is very grateful for these insightful comments.
The author has revised the article according to the requirements of the reviewer.
The response letter has been uploaded to the attachment.

Reviewer 2 Report
In this paper,the milling force prediction model considering Taylor factor is proposed, and many experiments are carried out to obtain milling force, and the micro-structure evolution of milling surface before and after milling is observed by EBSD. Then, the reliability of the prediction model proposed is validated. This paper is rich in content, but it needs a major revision according to the following comments:
1. Is the grain size related to the milling prediction model?
2. The relation of milling force prediction model and microstructure evolution is vague and should add some description.
3. The analysis of experimental results is not deep enough.
4. Some grammatical and word errors need to be corrected.
Author Response

(The authors gave the same response as above.)

Reviewer 3 Report
The paper deals with the TC4 Milling Force Prediction Model based on Taylor Factor Model and Microstructure Evolution of Milling Surface.
According to the reviewer, the paper is worth publishing at micromachines Journal, since corrections are needed and then the paper can be accepted for publication in the journal.
While the authors have made considerable research effort, the presentation of the paper and the results must be proved. Additionally make the following corrections to the manuscript:
Comment 1
Line 6
The authors must add the Country.
Comment 2
The authors must format the paper according to the journal's instructions:
Line 50
Merchant and Oxley et al [7, 8],
The authors must replace
Merchant [7] and Oxley [8],
Lines 52 - 53
Endres [9] and Moufki
[10]
The authors must replace
Endres et al. [9] and Moufki et al. [10]
Line 58
Zhou [12] et al proposed
The authors must replace
Zhou et al. [12] proposed
Line 61
Kang [13] et al considered
The authors must replace
Kang [13] considered
Line 64
Peng [14] et al established
The authors must replace
Peng [14] established
Line 100
by Ji [15] et al,
The authors must replace
by Ji [15],
The authors must replace the Fig. to Figure (for the text).
Line 151
Zhao [23, 24]
The authors must replace
Mao et al. [23] and Zhao [24]
Line 209
Ji [15] et al established
The authors must replace
Ji [15] established
Line 239
by Waldorf [34]
The authors must replace
by Waldorf et al. [34]
Line 314
by Ji [15] and Li [43] through
The authors must replace
by Ji [15] and Li et al. [43] through
Comment 3
Lines 104 and 105
The authors must give more details for experiment equipments (Kistler dynamometer, EBSD: type, model).
Comment 4
The authors must give more details for Table 1 (data from supplier or author's experiments?).
Comment 5
Lines 126 - 127
In order to investigate the evolution of micro-texture after a hot compression experiment,
The authors must explain what they mean with "the hot compression experiment".
Comment 6
First the authors mention Table 2 in the text and then Figure 2,
while the order presented in the paper is first Figure 2 and then Table 2.
Move the Figure 2 before Table 2.
Comment 7
Tables 6 - 8
The authors must insert a space between symbols.
For example:
?(???)
? (?P?)
Comment 8
Table 9
The authors must explain why sometimes the forces ft and fn are greater than Pred Ft and Pred Fn respectively, and sometimes they are smaller.
Comment 9
Major Problem
The authors must explain (with more details) how the Tables 10 and 11 occur.
The authors must explain the values 14.2 - 20.7 - 14.3 - 12.3 - 12.9 as a Relative Error (%) (Table 12)
Comment 10
The Figure 10 must be accompanied on the same page as the Figure's title.
Comment 11
Changes the References section format.
According to the journal's instructions:
1. Author 1, A.B.; Author 2, C.D. Title of the article. Abbreviated Journal Name Year, Volume, page range.
The authors must correct the References according to the journal's instructions.
The authors must remove the symbols [J] [D] and [M].
Increase the number of the reference papers including (primarily) from Micromachines.
The authors use 0 paper from Micromachines journal / 0 papers from MDPI Journals / 43 papers from journals (References)
Τhe number for papers from MDPI journals
is considered insufficient (in reviewer's opinion).
Author Response

(The authors gave the same response as above.)

Round 2
Reviewer 1 Report
Please include the idea of cryogenics with CO2 in the possible techniques, by O. Pereira.
Reviewer 2 Report
No
Reviewer 3 Report
Comment 1
Line 61
Kim [13] carried out
The authors must replace
Kim et al. [13] carried out
Comment 2
Line 619
Plastic strain in metals [J].
The authors must delete the "[J].
Comment 3
The authors must increase the quality of the presentation.
The Figures and the Tables must be accompanied on the same page as the Figures or Tables title (if it is possible).